# Pay Attention to Real World Perturbations! Natural Robustness Evaluation in Machine Reading Comprehension

## Abstract

As neural language models achieve human-comparable performance on Machine Reading Comprehension (MRC) and see widespread adoption, ensuring their robustness in real-world scenarios has become increasingly important. Current robustness evaluation research, though, primarily develops synthetic perturbation methods, leaving unclear how well they reflect real life scenarios. Considering this, we present a framework to automatically examine MRC models on naturally occurring textual perturbations, by replacing paragraph in MRC benchmarks with their counterparts based on available Wikipedia edit history. Such perturbation type is *natural* as its design does not stem from an arteficial generative process, inherently distinct from the previously investigated synthetic approaches. In a large-scale study encompassing SQUAD datasets and various model architectures we observe that natural perturbations result in performance degradation in pre-trained encoder language models. More worryingly, these state-of-the-art Flan-T5 and Large Language Models (LLMs) inherit these errors. Further experiments demonstrate that our findings generalise to natural perturbations found in other more challenging MRC benchmarks. In an effort to mitigate these errors, we show that it is possible to improve the robustness to natural perturbations by training on naturally or synthetically perturbed examples, though a noticeable gap still remains compared to performance on unperturbed data.

## 1 Introduction

Transformer-based pre-trained language models demonstrate remarkable efficacy in addressing questions based on a given passage of text, a task commonly referred to as Machine Reading Comprehension (MRC) (Devlin et al., 2019; Brown et al., 2020; He et al., 2021; Wei et al., 2022; Touvron et al., 2023; OpenAI et al., 2024). Despite these advancements, high-performing MRC systems are also known to succeed by relying on shortcuts in benchmark datasets rather than truly demonstrating understanding of the passage, thereby lacking robustness to various types of test-time perturbations (Ho et al., 2023; Schlegel et al., 2023; Levy et al., 2023).

Evaluating models' resilience to textual perturbations during inference aids in identifying adversarial instances that highlight their shortcut behaviour and provides insights into mitigating these shortcuts (Ho et al., 2023). While numerous synthetic perturbation approaches have been explored and reveal the vulnerabilities of MRC models to various linguistic challenges (Ribeiro et al., 2018; Jiang & Bansal, 2019; Welbl et al., 2020; Tan et al., 2020; Tan & Joty, 2021; Schlegel et al., 2021; Cao et al., 2022; Tran et al., 2023), a serious concern is that these carefully designed perturbations might not necessarily appear in real-world settings. Consequently, this poses a risk of neglecting the weaknesses of reading comprehension systems to real challenges when deployed in practical scenarios, thus potentially hindering the improvement of their reliability in practical applications.

To counteract this issue, in this paper, we develop a framework to inject textual changes that arise in real-world conditions into MRC datasets and audit how well contemporary language models perform under such perturbations. We deem them as *natural* because the perturbation process does not involve any artificial manipulation, in line with the definitions by Belinkov & Bisk (2018); Hendrycks et al. (2021); Pedraza et al. (2022); Agarwal et al. (2022); Le et al. (2022) (Figure 1).

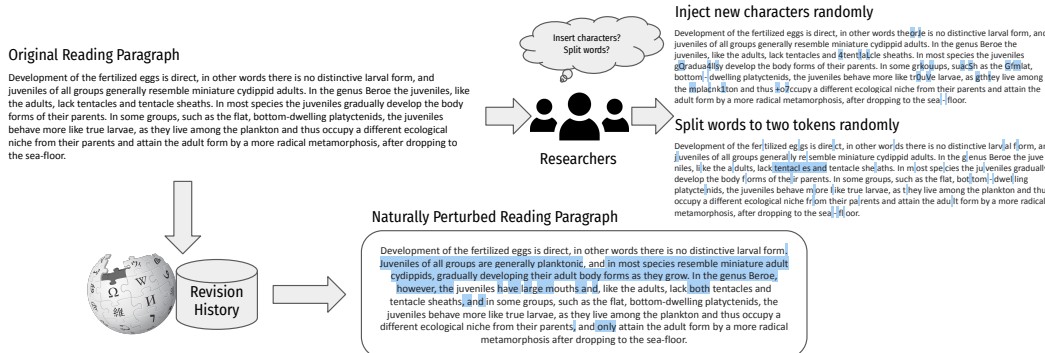

Figure 1: Given a reading context, we extract and use Wikipedia revision history to construct its naturally perturbed version for a more realistic robustness evaluation (Bottom), rather than relying on a set of synthetic methods (Top).

Results of robustness evaluation are therefore more representative of real-world applications. Similar to Belinkov & Bisk (2018), our approach utilises Wikipedia revision histories as the source of natural perturbations, given that the differences between revisions authentically capture the textual modifications made by human editors in the real world[1]. Despite this, significant differences exist in the perturbation construction methodology between us. Perturbation in (Belinkov & Bisk, 2018) is restricted to single word replacements and applied on non-English source-side sentences in machine translation. In detail, they build a look-up table of possible lexical replacements by harvesting naturally occurring errors (typos, misspellings, etc.) from available corpora of French/German Wikipedia edits (Max & Wisniewski, 2010; Zesch, 2012). Afterwards, they replace every word in the source-side sentences with an error if one exists in the look-up table. Different from (Belinkov & Bisk, 2018), our approach does not restrict the perturbation level and utilise English Wikipedia. By comparing the variances between each adjacent revision, we identify perturbed versions for each Wikipedia reading passage in the original MRC benchmarks (if it exists). This enables us to capture more comprehensive and critical natural perturbation patterns (see Section 5.2) that can not be possible to capture in (Belinkov & Bisk, 2018). Our perturbation method only alter the reading context, while the questions and ground truth answers remain unchanged.

With the established framework, we conduct extensive experiments on five datasets, evaluating twenty-nine models, including nine recently proposed LLMs. Experimental results on Stanford Question Answering Dataset (SQUAD) (Rajpurkar et al., 2016; 2018) indicate that natural perturbations encompass rich linguistic variations and can lead to failures in the encoder-only models, while humans are almost undeterred by their presence. Crucially, these errors also transfer to larger and more powerful models, such as `Flan-T5` and state-of-the-art LLMs. These findings also generalise to other and more challenging MRC benchmark (e.g., HOTPOTQA (Yang et al., 2018)) resulting in a decrease of SOTA LLMs' performance, emphasising its harmful effects. Adversarial retraining with either naturally or synthetically perturbed MRC instances can enhance the robustness of encoder-only models against natural perturbations, with the latter sometimes providing greater benefits. However, there is still ample room for improvement, calling for better defense strategies.

The contributions of this paper are as follows:

- A novel Wikipedia revision history-based framework to generate *natural* perturbed MRC benchmarks for *realistic* robustness evaluation.

- Empirical demonstration of the validity of natural perturbations, their characterisation by different linguistic phenomena and their harmful effects on diverse model architectures across five benchmarks generated with the proposed framework.

---

[1] Wikipedia happens to allow us to track changes and automatically construct a benchmark to test the behaviour of neural language models on natural perturbations, but the phenomenon of natural perturbations is by no means limited to Wikipedia. Instead, these can occur in any kind of text that evolves over time.

- Showcasing adversarial re-training with natural or, especially, synthetic perturbations as a way to enhance the robustness of encoder-only MRC models against natural perturbations.

## 2 RELATED WORK

**Robustness Evaluation in MRC**   A typical approach to evaluate the robustness of MRC models is via test-time perturbation. This line of research develops different perturbation methods as attacks, such as adversarial distracting sentence addition (Jia & Liang, 2017; Tran et al., 2023), low-level attacks (Eger & Benz, 2020), word substitution (Wu et al., 2021), character swap (Si et al., 2021), entity renaming (Yan et al., 2022) and paraphrasing (Gan & Ng, 2019; Lai et al., 2021; Wu et al., 2023). Our work also fits within the category of test-time perturbation, but differs from previous works in that we introduce perturbations that naturally occur in real-world scenarios, therefore contributing to a more practical robustness examination.

**Natural Perturbation for Robustness Assessment**   Compared with deliberately crafting the perturbed instances, the study of natural perturbation is under-explored. In the computer vision domain, researchers find that real-world clean images without intentional modifications can confuse deep learning models as well, terming them as natural adversarial examples (Hendrycks et al., 2021; Pedraza et al., 2022). Similarly, in the field of Natural Language Processing (NLP), naturally occurring perturbations extracted from human-written texts can also degrade model performance in tasks such as machine translation (Belinkov & Bisk, 2018) and toxic comments detection (Le et al., 2022). Motivated by these, we attempt to harvest natural perturbations from available Wikipedia revision histories and utilise them to modify the original MRC instances. To the best of our knowledge, we are the first to investigate MRC model robustness under real natural perturbations. Furthermore, it should be noted that the concept of natural perturbed examples in this paper differs from what is defined in previous NLP literature, where the latter measures the extent to which synthetically modified text preserves certain linguistic characteristics such as fluency, coherence, grammaticality and clarity, i.e., its naturalness (Jin et al., 2020; Li et al., 2020; Schlegel et al., 2021; Qi et al., 2021; Wang et al., 2022a; Dyrmishi et al., 2023). Some works also propose that a natural synthetically perturbed sample should be imperceptible to human judges (Li et al., 2020; Garg & Ramakrishnan, 2020) or convey the impression of human authorship (Dyrmishi et al., 2023). However, this proposition remains a subject of debate (Zhao et al., 2018; Wang et al., 2022b; Chen et al., 2022).

## 3 NATURAL PERTURBATION PIPELINE

We design a pipeline to automatically construct label-preserving stress MRC test sets with noises that occur in real-world settings by leveraging Wikipedia revision histories (Figure 2). Our approach comprises two modules: *candidate passage pairs curation* and *perturbed test set construction*.

**Candidate passage pairs curation.**   For each English Wikipedia article within the development set[2] of MRC datasets, we systematically extract its entire revision histories and preprocess them, including the removal of markups and the segmentation of content. Subsequently, we obtain the content differences between each current revision and the previous adjacent one, identifying three distinct editing patterns: addition, deletion, and modification. In the case of an edit falling within the modification pattern, we retain the paragraph from the prior version as the *original* and the corresponding one from the current version as the *perturbed*, provided both paragraphs exceed 500 characters[3].

**Perturbed test set construction.**   To generate the naturally perturbed test set, we begin by acquiring all reading passages from the development set of each MRC dataset and identifying their entries in the collection of previously extracted candidate original passages, along with the corresponding perturbed counterparts. Subsequently, for the matched original passages with a single occurrence,

---

[2]Since not all test sets are public, we apply natural perturbations to the development sets. For simplicity, we use the term "test set" throughout.

[3]This threshold setting adheres to the methodology employed in the collection of SQuAD 1.1 (Rajpurkar et al., 2016).

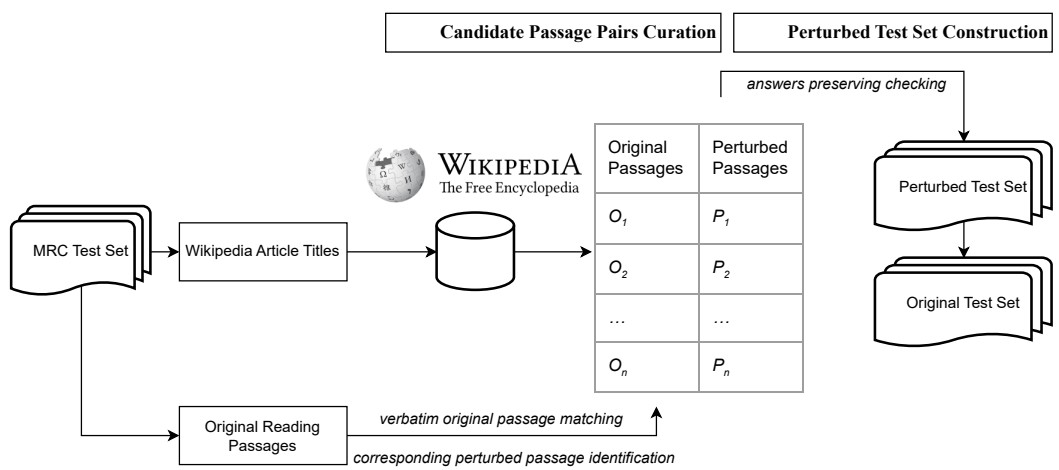

Figure 2: Process of generating naturally perturbed MRC test sets.

we keep them and the corresponding perturbed passages; whereas for those with multiple occurrences, we randomly select one instance for each and extract its perturbed version. After obtaining the perturbed reading passages, we retain only those with at least one question where all annotated ground truth answers (or all plausible answers for the unanswerable question) can still be located within the perturbed context, resulting in the *Perturbed* test set. For the sake of comparison, we also construct an *Original* version of the test set keeping only the original passages and questions corresponding to those that were included in the *Perturbed* version.

# 4 EXPERIMENT SETUP

## 4.1 DATASETS

We select five MRC datasets: SQUAD 1.1 (Rajpurkar et al., 2016), SQUAD 2.0 (Rajpurkar et al., 2018), DROP (Dua et al., 2019), HOTPOTQA (Yang et al., 2018) and BOOLQ (Clark et al., 2019). These are chosen due to the fact that their reading passages are sourced from Wikipedia, thereby enabling the utilisation of Wikipedia editing histories to generate the naturally perturbed test set.

## 4.2 MODELS

Our evaluation study involves multiple contemporary MRC models across three different types: encoder-only, encoder-decoder, and decoder-only. Under the encoder-decoder and decoder-only model evaluation settings, we reframe the extractive MRC as the text generation task based on the given context and question. Access to and experimentation with all models are possible via the use of the HuggingFace's *Transformers* library (Wolf et al., 2020), two 80GB Nvidia A100 GPUs and the OpenAI ChatGPT API.

**Encoder-only:** We select BERT (Devlin et al., 2019) and its various variants for evaluation, including DistilBERT (Sanh et al., 2019), SpanBERT (Joshi et al., 2020), RoBERTa (Liu et al., 2019), ALBERT (Lan et al., 2020) and DeBERTa (He et al., 2021). Some of these model types also come with different variations, such as size (e.g., *base* and *large* for RoBERTa), versions (e.g., *v1* and *v2* for ALBERT) and whether the input text is cased or not (e.g., *cased* and *uncased* for BERT), all of which are included in the evaluation. We fine-tune these encoder-only pre-trained language models on the training set of the two SQUAD datasets (Rajpurkar et al., 2016; 2018) and evaluate them on the constructed original and perturbed test sets. Model details and the hyperparameters used in model fine-tuning are shown in Appendix A.

**Encoder–Decoder:** Instruction finetuning has been demonstrated to be effective in enhancing zero-shot performance of pretrained language models, resulting in the development of Finetuned

Language Net (FLAN) (Wei et al., 2022). In this work, we use the instruction-finetuned version of T5 model class, specifically the `Flan-T5` (Chung et al., 2022), available in sizes ranging from *small* (80M), *base* (250M), *large* (780M) to *xl* (3B). During evaluation, we utilise the instruction templates from MRC task collection in open-sourced FLAN repository and report the model performance as the average of those obtained across the employed templates. Refer to Appendix B for various instruction templates used for the evaluation on the test sets with the format as the two SQuAD datasets.

**Decoder-only:** There is an exponential increase of pre-trained generative LLMs and their fine-tuned chat versions, inspired by the remarkable success of ChatGPT (Bang et al., 2023). Therefore, our experiments incorporate a broad range of recently proposed language model families, including `GPT 3.5 Turbo`, `GPT-4o`, `Llama 2` (Touvron et al., 2023), `Llama 3` (Dubey et al., 2024), `Mistral` (Jiang et al., 2023), `Falcon` (Almazrouei et al., 2023) and `Gemma` (Mesnard et al., 2024). The zero-shot prompts designed for soliciting their responses are presented in Appendix C.

### 4.3 MODEL EVALUATION METRICS

In line with existing literature, we choose the Exact Match (EM) and (instance-averaged) Token-F1 score to assess the performance of both encoder-only and encoder-decoder models (Rajpurkar et al., 2016), as on SQuAD-style test sets, they are optimised to output the shortest continuous span from the context as the answer (or predict the question as unanswerable) during inference. However, the outputs of the decoder-only models do not consistently adhere to the instruction due to their conversational style, rendering EM and F1 unsuitable for evaluation. Consequently, we employ a more lenient metric, namely Inclusion Match (IM), which measures whether the response of the model contains any of the ground truth answers (Bhuiya et al., 2024). Furthermore, if the model's output includes phrases such as "I cannot answer this/the question" or "unanswerable"[4], we deem that the model believes the question is not answerable. Model robustness is quantified by measuring the relative variation in performance (as reflected in the F1 or IM) under perturbations.

## 5 MRC UNDER NATURAL PERTURBATION

In this section, we present the main findings of our study. Our intention in starting with SQuAD and encoder-only models is to establish a baseline evaluation of model behaviour under natural perturbations. While SQuAD is less challenging, its simplicity enables a focused and controlled examination of the perturbation effects (Section 5.1), error sources (Section 5.2) and adversarial instance validity (Section 5.3), providing a foundation for generalising our findings to more complex datasets and model architectures. As we observe that encoder-only models suffer from natural perturbations on the SQuAD datasets, we further investigate the transferability of the errors generated by encoder-only models to other model architectures (Section 5.4)[5]. We finally show that the behaviour we observe in the baseline evaluation (i.e., encoder-only models suffer from natural perturbations on the SQuAD datasets) also carries over to more powerful LLMs and other more complex datasets (Section 5.5).

### 5.1 ARE ENCODER-ONLY MRC MODELS RESILIENT TO NATURAL PERTURBATION?

Table 1 presents the relative F1 change for all encoder-only MRC models on the naturally perturbed test set generated based on the SQuAD 1.1 and SQuAD 2.0 development set, respectively. It can be clearly seen from Table 1 that overall, the performance of all the examined models decreases, indicating that ***encoder-only MRC models suffer from natural perturbation***. However, we notice that the performance drop of all models is negligible (the biggest drop is only 3.06%), which suggests that those models also exhibit considerable robustness to natural perturbations.

---

[4]We collate a collection of such phrases by manually examining the decoder-only models' outputs (Check Appendix D for the full set).

[5]Alternatively, we also study the effect of all perturbed instances on each architecture type and measure the transferability of adversarial examples across all models (see Appendix E).

Table 1: Relative F1 change (%) for encoder-only MRC systems subjecting to natural perturbations. For SQuAD 2.0, the overall values are broken down to answerable and unanswerable questions, respectively.

| Victim | SQuAD 1.1 | SQuAD 2.0 | |
|---|---|---|---|
| | | Overall | (Ans./Unans.) |
| distilbert-base | −0.6 | −0.71 | (−2.76/1.71) |
| bert-base-cased | −0.21 | −0.63 | (−1.84/0.6) |
| bert-base-uncased | −0.87 | −0.49 | (−1.88/0.94) |
| bert-large-cased | −0.63 | −0.53 | (−1.61/0.55) |
| bert-large-uncased | −0.35 | −1.38 | (−2.51/−0.24) |
| spanbert-base-cased | −0.26 | −1.24 | (−2.66/0.15) |
| spanbert-large-cased | −0.51 | −1.20 | (−1.9/−0.56) |
| roberta-base | −0.61 | −0.60 | (−2.09/0.81) |
| roberta-large | −0.29 | −1.52 | (−2.6/−0.54) |
| albert-base-v1 | −1.0 | −1.07 | (−2.02/−0.22) |
| albert-base-v2 | −0.34 | −1.08 | (−2.03/−0.22) |
| albert-large-v1 | −0.42 | −0.41 | (−1.42/0.52) |
| albert-large-v2 | −0.8 | −0.69 | (−1.66/0.22) |
| albert-xxlarge-v1 | −0.75 | −1.23 | (−3.06/0.49) |
| albert-xxlarge-v2 | −0.46 | −1.28 | (−3.02/0.36) |
| deberta-large | −0.52 | −1.05 | (−2.2/0.0) |

## 5.2 ERROR ANALYSIS

Although encoder-only MRC models exhibit a relatively small performance gap, it remains worthwhile to investigate the sources of natural perturbation and reveal the perturbation phenomena contributing to models' error. To this end, we manually label linguistic features between passages where models succeed and fail, to identify how they differ.

Within the original and the naturally perturbed test set pair generated based on SQuAD 2.0 development set, we first identify 384 instances where at least one encoder-only model succeeds on the original but fails[6] on the perturbed (i.e., being adversarial), and then randomly select the same number of instances on which all encoder-only models succeed on both the original and perturbed versions (Naik et al., 2018). We refer to these two types of instances as C2W (correct to wrong) and C2C (correct to correct) instances, respectively. Among the identified C2W and C2C instances, we further remove duplicates, resulting in 210 and 244 unique original and perturbed paragraph pairs, respectively. Furthermore, as natural perturbation can occasionally help the model to get the answer correct, we also filter 85 unique W2C (wrong to correct) instances on which at least two encoder-only models fail on the original but succeed on the perturbed. Finally, utilising an 8-category taxonomy of the semantic edit in-

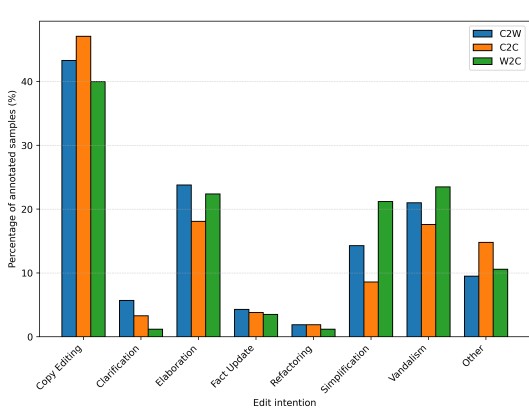

Figure 3: The percentage (%) of samples annotated with each edit intention in the C2W, C2C and W2C categories. The percentages do not add up to 100% because a single revision may fall into multiple intentions.

tentions in Wikipedia revisions derived from Yang et al. (2017), the chosen 210 samples of C2W and C2C, as well as the 85 W2C were annotated, with 20% of the annotated C2W and C2C examples presented to a second annotator for additional validation. See Appendix F for the instruction provided to the annotators, along with detailed explanations of each edit intention. We calculate the

---

[6]For answerable questions, a model's prediction is considered correct if EM score equals 1, and incorrect if F1 score is 0 or it determines the question is unanswerable. For unanswerable questions, a model's prediction is correct if it predicts the question is unanswerable, and wrong if it provides an answer span.

(micro-averaged) F1 score to evaluate the inter-annotator agreement, which is $0.82$. This suggests that the annotators' annotations align closely. Figure 3 reports the annotation results.

Distribution of perturbation types shown in Figure 3 generally aligns with the edit intentions distribution annotated in (Yang et al., 2017), with *Copy Editing* and *Elaboration* appearing more frequently than others, such as *Clarification*, *Fact Update*, and *Refactoring* [7]. From Figure 3, we observe that there is no significant difference in the distribution of annotated edit intentions between C2W and C2C examples, suggesting that ***though these types of natural perturbations confuse the encoder-only MRC models, there seems no correlation with human-perceivable features***. A roughly similar distribution is also observed in the W2C examples, which indicates that these natural perturbation types can also facilitate correct answers by the models, i.e., being beneficial. These demonstrate that on SQUAD 2.0, there might be no correlation between the quality of the naturally perturbed passage and its potential for being adversarial [8]. Certain text edits aimed at improving the passage quality, such as *Copy Editing* and *Elaboration*, do render the perturbation adversarial, whereas edits intended to damage the article may not consistently result in adversarial instances; in fact, *vandalism* can even assist models in providing correct answers. Instead, we infer that whether an edit to the passage can render the MRC instance adversarial or not depends on the location of the edits in relation to the question. Among the $384$ C2W and C2C examples, we measure the proportion of answerable questions with the answer sentence(s) in the original passage remaining unmodified in the naturally perturbed version, which is $34.5\%$ and $71.5\%$, respectively. This confirms our hypothesis that if the edits affect the answer sentence(s), there is a higher likelihood of the perturbed example becoming adversarial; otherwise, it might not. *Copy Editing appears to alter the answer sentences in the reading passage more frequently, making it the most impactful category that confuses models (contributing to more than 40% of error cases), while other types have a lesser effect.* Appendix G presents one perturbed example for each of the C2W, C2C, and W2C categories, respectively, along with the annotated natural perturbation type(s).

## 5.3 VALIDITY OF NATURE ADVERSARIAL EXAMPLES

To accurately assess a model's robustness under perturbation, it is vital to examine the validity of adversarial example, i.e. whether humans can still find the correct answer under the perturbation (Dyrmishi et al., 2023). We first present two human annotators with the same collection of adversarial instances, which includes only perturbed contexts and their corresponding questions, and then ask them to answer the question based on the perturbed context. The annotators are required to select the shortest continuous span in the perturbed context that answers the question and are allowed to leave the answer blank if they are confident that the question is not answerable. Full instructions given to the annotators can be seen in Appendix F. Subsequently, for both annotators, we measure the correctness (1 or 0) of their provided answers by comparing each of them with the corresponding ground truth answers[9]. The inter-annotator agreement is then measured by computing the Cohen's $\kappa$ coefficient (Cohen, 1960). We then involve a third human annotator to annotate the adversarial examples on which the first two annotators disagree and then take the majority label as ground truth.

We employ this approach to verify the validity of the $210$ C2W examples in Section 5.2 and find that $86\%$ of these adversarial examples are valid ($0.77$ Cohen's $\kappa$), indicating that ***a substantial proportion of natural adversarial examples for encoder-only MRC model(s) are valid***.

## 5.4 CAN ERRORS FROM ENCODER-ONLY MODELS AFFECT OTHER ARCHITECTURES?

We further investigate whether the errors identified in encoder-only models carry over to other more recent models and architectures, as state-of-the-art advancements in NLP would suggest otherwise. Therefore, we propose an exhaustive search algorithm that leverages the predictions of all encoder-only models to create the challenging natural perturbed test set. In detailed terms, for each matched reading passage from the prior version and its counterpart from the current version, we determine

---

[7]This reflects the inherent characteristics of Wikipedia revisions.

[8]We also find little or no significant correlation between the perturbation magnitude (measured as byte-level changes between the original and perturbed passages) and model failure, with point biserial correlation coefficient close to 0.

[9]Here, as long as one of the ground truth answers is included in the human-provided answer span, we consider the prediction to be correct.

which should be designated as the *original* and which as the *perturbed* based on which scenario can yield the questions on which the maximum sum of the number of encoder-only models demonstrates the lack of robustness phenomenon[10]. Questions on which none of the encoder-only models fail under the perturbation are then removed. A more detailed explanation of this process is provided in Appendix H. We finally process the identified original and perturbed passage pairs to ensure that the original passages are within the original SQUAD 1.1 development set. For those original passages with multiple occurrences, we select the one with the maximum number of questions reserved.

With the development set of SQUAD 1.1 and SQUAD 2.0 as the source, this results in two challenge perturbed test sets: NAT_V1_CHALLENGE and NAT_V2_CHALLENGE. In NAT_V1_CHALLENGE, there are 184 contexts and 234 questions. NAT_V2_CHALLENGE contains 214 contexts and 442 questions (226 unanswerable).

Table 2 shows the evaluation results of both encoder-decoder and decoder-only models on the newly generated challenge test sets. From the table, we observe that ***the errors caused by natural perturbation in encoder-only MRC models transfer to both `Flan-T5` and LLMs***. On the NAT_V1_CHALLENGE, `Flan-T5-small` demonstrates the greatest susceptibility to natural perturbation, experiencing a 14.27% decrease in F1, while among LLMs, `Gemma-7B-IT` emerges as the least robust, with a 16.66% IM drop. Transitioning to the NAT_V2_CHALLENGE, the base version of `Flan-T5` exhibits the largest performance decline (13.83%) and `Falcon-7B-Instruct` stands out as the LLM with the lowest robustness. Further, the robustness of models under natural perturbations does not necessarily correlate with their size. For example, on NAT_V2_CHALLENGE, `Llama 2-chat-7B` demonstrates higher overall robustness than `Llama 2-chat-13B`, while `flan-t5-xl` exhibits the largest performance decrease (12.79%) compared to its small and large versions. In Appendix I, we showcase two adversarial examples targeting LLMs sourced from our generated challenge sets.

Table 2: The performance (%) of encoder-decoder and decoder-only MRC models on the newly generated original and naturally perturbed challenge test sets. Values in smaller font are changes (%) relative to the original performance of the model.

| Model | Performance | | | |
|-------|-------------|---|---|---|
| | *original* **vs.** *perturbed* | | | |
| | NAT_V1_CHALLENGE | | NAT_V2_CHALLENGE | |
| flan-t5-small | 58.76/64.76 | $48.58/55.52_{-14.27}$ | 42.57/44.57 | $39.71/41.81_{-6.19}$ |
| flan-t5-base | 79.49/85.01 | $66.1/73.42_{-13.63}$ | 70.66/72.85 | $61.16/62.78_{-13.83}$ |
| flan-t5-large | 88.1/92.53 | $76.57/82.31_{-11.05}$ | 79.11/81.01 | $70.14/72.13_{-10.96}$ |
| flan-t5-xl | 86.25/91.57 | $75.0/81.45_{-11.05}$ | 83.71/85.84 | $73.19/74.86_{-12.79}$ |
| GPT-3.5-turbo-0125 | 91.03 | $83.33_{-8.46}$ | 51.58 | $47.06_{-8.76}$ |
| GPT-4o-2024-08-06 | 94.87 | $82.48_{-13.06}$ | 82.81 | $71.72_{-13.39}$ |
| Gemma-2B-IT | 51.28 | $43.16_{-15.83}$ | 55.66 | $50.23_{-9.76}$ |
| Gemma-7B-IT | 82.05 | $68.38_{-16.66}$ | 59.95 | $57.01_{-4.9}$ |
| Llama 2-chat-7B | 82.91 | $73.93_{-10.83}$ | 41.63 | $38.69_{-7.06}$ |
| Llama 2-chat-13B | 80.77 | $73.93_{-8.47}$ | 46.83 | $41.18_{-12.06}$ |
| Llama-3-8B-Instruct | 88.89 | $77.35_{-12.98}$ | 51.81 | $46.61_{-10.04}$ |
| Mistral-7B-Instruct-v0.2 | 85.9 | $76.92_{-10.45}$ | 55.43 | $52.04_{-6.12}$ |
| Falcon-7B-Instruct | 53.42 | $50.00_{-6.4}$ | 32.81 | $23.53_{-28.28}$ |
| Falcon-40B-Instruct | 69.66 | $62.82_{-9.82}$ | 38.69 | $36.88_{-4.68}$ |

## 5.5 DO OUR FINDINGS GENERALISE TO OTHER MRC DATASETS?

The two SQUAD datasets investigated previously are relatively simple, as they lack challenging features (Schlegel et al., 2020), leading to super-human performance of MRC models (Lan et al., 2020). To generalise our findings to more challenging MRC benchmarks, we apply the natural perturbation methodology (Section 3) to the development set of three more datasets and assess the performance changes of several LLMs. For DROP (Dua et al., 2019), we first use the `GPT-4o`

---

[10]We define A model as lacking robustness to the perturbation if it achieves 1 EM on the original question but attains less than 0.4 F1 on the perturbed one (for answerable questions).

mini to infer the likely Wikipedia article title from which each passage is retrieved[11] and extract the revision histories for 224 out of 473 articles. For HOTPOTQA (Yang et al., 2018), we use its development set in the "distractor" setting and extract revision histories for around 8.4% (1156) Wikipedia articles containing the supporting facts. For BOOLQ (Clark et al., 2019), we extract revision histories for around 19.4% (514) Wikipedia articles.

Overall, as shown in Table 3, we find that *when natural perturbations are applied to more challenging benchmarks, LLMs also exhibit a lack of robustness*. This emphasises the broadly negative impact of natural perturbations. On naturally perturbed HOTPOTQA, these models exhibit the largest average IM drop (5.99%), suggesting that natural perturbations significantly destroy the multi-hop reasoning chain. Furthermore, natural perturbations can also not harm or even benefit some models in answering questions on certain benchmarks, possibly due to the characteristics of those benchmarks. We leave this investigation for future work.

Table 3: IM changes (%) of LLMs on naturally perturbed test sets of three challenging MRC datasets.

| LLM | IM Relative Change (%) | | |
|---|---|---|---|
| | DROP | HOTPOTQA | BOOLQ |
| Gemma-2B-IT | −19.44 | - | −8.01 |
| Gemma-7B-IT | −6.01 | −6.45 | - |
| Llama 2-chat-7B | −1.89 | −4.92 | 8.69 |
| Llama 2-chat-13B | −1.89 | −4.41 | −1.81 |
| Llama-3-8B-Instruct | −4.69 | −3.33 | 3.45 |
| Mistral-7B-Instruct-v0.2 | −5.01 | −4.22 | 3.51 |
| Falcon-7B-Instruct | −6.04 | −15.38 | 2.22 |
| Falcon-40B-Instruct | 7.88 | −12.52 | −9.8 |
| GPT-4o-2024-08-06 | −12.68 | −2.67 | −7.47 |
| *average* | −5.53 | −5.99 | −1.02 |

## 6 DEALING WITH NATURAL PERTURBATIONS

In this section, we provide an initial exploration of methods to defend against natural perturbations, focusing on encoder-only models and SQuAD datasets. Expanding to other datasets and architectures could be explored in future work. To enhance model robustness, we conduct adversarial training by identifying six encoder-only model architectures that already exhibit the highest robustness to natural perturbations in their respective categories (except albert-xxlarge-v2 on NAT_V2_CHALLENGE), and presenting them with both original training data and the generated naturally perturbed training examples. We extract the entire Wikipedia revision histories for the 392 articles in the original SQUAD training set, and then obtain 5,262 (with 22,033 questions) and 5,311 (with 32,993 questions) perturbed contexts to augment the original SQUAD 1.1 and SQUAD 2.0 training set, respectively, using the methodology described in Section 3. Table 4 compares the performance of these models on NAT_V1_CHALLENGE and NAT_V2_CHALLENGE, before and after retraining.

Apart from re-training with the same type of noise, we also ask whether exposing models to synthetic perturbations can help them confront natural ones. Therefore, we incorporate thirteen synthetic perturbation techniques spanning character and word levels (see Appendix J). Afterwards, we first retrain deberta-large with perturbed training samples generated by each synthetic perturbation method, respectively, and assess the performance changes compared to the vanilla version on both NAT_V1_CHALLENGE and NAT_V2_CHALLENGE (Figure 8 in Appendix K). As we observe that synthetic adversarial training can assist deberta-large in handling natural perturbations, we further retrain five other models in the same manner and quantify the performance difference on NAT_V1_CHALLENGE compared to the vanilla version, as shown in Figure 4.

In general, for encoder-only MRC models, retraining with natural perturbations enhances the performance on naturally perturbed test sets and improves the robustness to such perturbations as well, though this can lead to varying reductions in performance on the clean test set. Encouragingly, adversarial training with synthetically perturbed examples benefits the model's capability to handle natural perturbations as well, a phenomenon differs from what is reported in machine translation task (Belinkov & Bisk, 2018). In some cases, the improvement even exceeds what achieved by retraining the model on natural perturbations alone. We also observe that the effectiveness of adversarial training varies with model size and architecture. Generally, adversarial training brings the most significant benefits for the weakest distilbert-base, with the benefits diminishing in larger and more complex model architectures.

---

[11]Prompt: *"Given a reading paragraph, return the Wikipedia page title from which it is likely retrieved."*

Table 4: Comparison of the performance of several encoder-only MRC systems on NAT_V1_CHALLENGE and NAT_V2_CHALLENGE, before and after re-training. The results shown in the shaded areas represent the performance of the model retrained on the augmented training set with naturally perturbed instances.

| Model | Performance (EM/F1) | | | |
| --- | --- | --- | --- | --- |
| | | *original* **vs.** *perturbed* | | |
| | NAT_V1_CHALLENGE | | NAT_V2_CHALLENGE | |
| distilbert-base | 64.53/70.45 | $41.03/47.6_{-32.43}$ | 56.56/59.08 | $41.18/43.3_{-26.71}$ |
| | 57.26/63.44 | $43.59/51.87_{-18.24}$ | 53.17/55.4 | $43.89/45.51_{-17.85}$ |
| bert-large-cased | 79.06/83.66 | $63.68/70.23_{-16.05}$ | 66.29/68.35 | $53.17/55.04_{-19.47}$ |
| | 74.79/80.14 | $59.83/67.5_{-15.77}$ | 67.87/69.31 | $58.37/59.53_{-14.11}$ |
| spanbert-large-cased | 84.19/88.2 | $67.95/74.77_{-15.23}$ | 78.73/80.68 | $62.44/64.99_{-19.45}$ |
| | 82.48/86.6 | $69.66/76.05_{-12.18}$ | 78.28/80.0 | $65.61/67.12_{-16.1}$ |
| roberta-large | 86.75/90.21 | $73.93/79.47_{-11.91}$ | 82.13/84.27 | $66.29/68.52_{-18.69}$ |
| | 83.33/87.15 | $70.94/76.53_{-12.19}$ | 81.22/82.67 | $70.59/71.84_{-13.1}$ |
| albert-xxlarge-v2 | 84.62/89.64 | $73.93/78.77_{-12.13}$ | 84.62/86.07 | $68.1/69.61_{-19.12}$ |
| | 86.32/90.93 | $75.64/81.07_{-10.84}$ | 82.58/84.08 | $70.59/72.78_{-13.44}$ |
| deberta-large | 88.46/92.5 | $73.5/78.48_{-15.16}$ | 85.07/86.65 | $71.49/73.0_{-15.75}$ |
| | 88.03/91.84 | $76.92/81.53_{-11.23}$ | 83.03/85.1 | $72.62/74.48_{-12.48}$ |

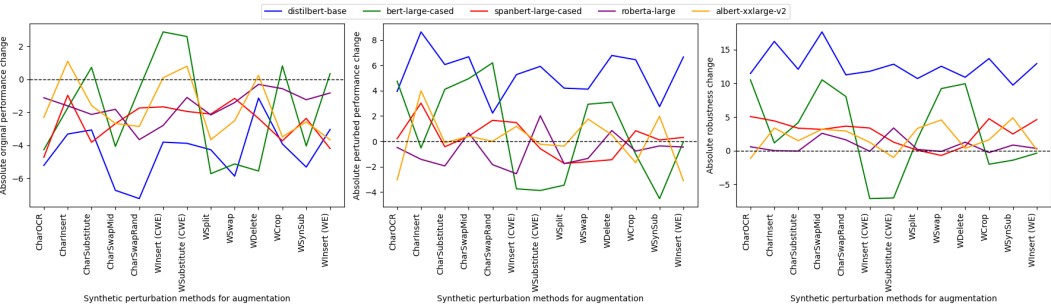

Figure 4: Absolute changes in original and perturbed performance (F1), as well as the robustness of five encoder-only models under natural perturbations (on NAT_V1_CHALLENGE), following re-training with each synthetic perturbation.

## 7 CONCLUSION

In this paper, we first study the robustness of MRC models to *natural* perturbations, which occur under real-world conditions without intentional human intervention. Using the proposed evaluation framework, we show that certain naturally perturbed examples can indeed be adversarial, i.e., lead to model failure, even when the modifications aim to improve the overall passage quality. Natural perturbations also appear to differ significantly from synthetic ones, exhibiting a wide range of rich linguistic phenomena and may be more effective in generating valid adversarial instances. Adversarial training via augmentation with either naturally or synthetically perturbed samples is generally beneficial for enhancing the model's robustness to natural perturbations; yet, it can decrease performance on clean test set. Future work includes the exploration of alternative natural perturbation approaches and the design of more effective defensive strategies against natural attacks.

ETHICS STATEMENT

All datasets, extracted natural perturbations, and models used in this work are publicly available. A very small proportion of natural perturbations may contain offensive content, as they come from reverted Wikipedia revisions intended to damage the articles. We include these to raise awareness within the community about their potential impact on MRC models and to call for methods to

improve the safety of MRC models–especially those LLMs operating under such adversarial conditions. Before starting the annotation task, we provide all annotators with clear instructions and inform the intended use of their annotations, obtaining their explicit consent. No private or sensitive information was collected, other than their annotations.

REPRODUCIBILITY STATEMENT

As part of the supplementary material, we release all our source code, along with the constructed naturally perturbed test sets and the augmented training sets with naturally or synthetically perturbed examples at `https://github.com/npanonymous/natural_perturbations`. We also provide the necessary information in models' evaluation setup, including the hyperparameters used to fine-tune and evaluate encoder-only models (Appendix A), prompt templates for evaluating `Flan-T5` (Appendix B) and other LLMs (Appendix C).

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

## A  ENCODER-ONLY MODEL PARAMETERS AND HYPERPARAMETERS FOR FINE-TUNING

Table 5 shows the hyperparameters used to fine-tune the pre-trained encoder-only MRC models in this work and their number of parameters contained.

Table 5: Number of parameters in each type of pre-trained encoder-only MRC model and the hyper-parameters used to fine-tune them. For BERT, SpanBERT, RoBERTa and ALBERT, we show the number of model parameters in the order of *base*, *large* and *xxlarge* (if applicable) version. d is the size of the token sequence fed into the model, b is the training batch size, lr is the learning rate, and ep is the number of training epochs. We used stride = 128 for documents longer than d tokens.

| $\text{Model}_{Parameters(M)}$ | d | b | lr | ep |
|---|---|---|---|---|
| $\text{DistilBERT}_{(66)}$ | 384 | 8 | $3e-5$ | 3 |
| $\text{BERT}_{(110/340)}$ | 384 | 8 | $3e-5$ | 2 |
| $\text{SpanBERT}_{(110/340)}$ | 512 | 4 | $2e-5$ | 4 |
| $\text{RoBERTa}_{(125/355)}$ | 384 | 8 | $3e-5$ | 2 |
| $\text{ALBERT}_{(11/17/223)}$ | 384 | 4 | $3e-5$ | 2 |
| $\text{DeBERTa}_{(350)}$ | 384 | 4 | $3e-6$ | 3 |

## B  INSTRUCTION TEMPLATES FOR FLAN-T5 EVALUATION

In Table 6, we present the instruction templates employed in constructing the inputs to the Flan-T5 model for the SQUAD 1.1 format and SQUAD 2.0 format test sets, respectively.

Table 6: Various instruction templates for Flan-T5 model evaluation.

| | SQUAD 1.1 |
|---|---|
| 1 | "Read this and answer the question\n\n{context}\n\n{question}" |
| 2 | "{context}\n{question}" |
| 3 | "Answer a question about this article:\n{context}\n{question}" |
| 4 | "Here is a question about this article: {context}\nWhat is the answer to this question: {question}" |
| 5 | "Article: {context}\n\nQuestion: {question}" |
| 6 | "Article: {context}\n\nNow answer this question: {question}" |

| | SQUAD 2.0 |
|---|---|
| 1 | "Read this and answer the question.  If the question is unanswerable, say \"unanswerable\".\n\n{context}\n\n{question}" |
| 2 | "{context}\n{question} (If the question is unanswerable, say \"unanswerable\")" |
| 3 | "{context}\nTry to answer this question if possible (otherwise reply \"unanswerable\"): {question}" |
| 4 | "{context}\nIf it is possible to answer this question, answer it for me (else, reply \"unanswerable\"): {question}" |
| 5 | "{context}\n\nAnswer this question, if possible (if impossible, reply \"unanswerable\"): {question}" |
| 6 | "Read this: {context}\nNow answer this question, if there is an answer (If it cannot be answered, return \"unanswerable\"): {question}" |

## C MRC Prompts

We use the following zero-shot prompts to instruct the decoder-only models to generate responses in the task of MRC.

**SQuAD 1.1**: *Use the provided article delimited by triple quotes to answer question. Provide only the shortest continuous span from the context without any additional explanation.* $\backslash n \backslash n$ *"""{context}"""* $\backslash n \backslash n$ *Question: {question}*

**SQuAD 2.0**: *Use the provided article delimited by triple quotes to answer question. Provide only the shortest continuous span from the context without any additional explanation. If the question is unanswerable, return "unanswerable".* $\backslash n \backslash n$ *"""{context}"""* $\backslash n \backslash n$ *Question: {question}*

**DROP & HotpotQA**: *Use the provided article delimited by triple quotes to answer question. Provide only the answer without any additional explanation.* $\backslash n \backslash n$ *"""{context}"""* $\backslash n \backslash n$ *Question: {question}*

**BoolQ**: *Use the provided article delimited by triple quotes to answer question. Return only TRUE or FALSE.* $\backslash n \backslash n$ *"""{context}"""* $\backslash n \backslash n$ *Question: {question}*

## D Indicators of Unanswerable

We manually identify a set of phrases contained in the output of LLMs that indicate the unanswerability of the question, including *"I cannot answer this/the question"*, *"unanswerable"*, *"There is no indication in the provided article"*, *"The context provided does not provide enough information"*, *"There is no reference in the given article"*, *"The answer to the question is not provided in the given article"*, *"it is not possible"*, *"question cannot be answered"* and *"context/question/article/text/article provided/passage does not"*.

## E Supplementary Experiments

We supplement Table 1 in Section 5.1 with additional experiments on `Flan-T5` and more recent LLMs such as `Gemma 2` (Team et al., 2024) and `Llama 3.2`, to study the effect of all perturbed instances on each architecture type. The results are presented in Table 7. From Table 7, we observe that similar to encoder-only models, `Flan-T5` and LLMs generally exhibit varying degrees of performance degradation under natural perturbations, but also exhibit considerable robustness to them.

Table 7: Performance change (%) for `Flan-T5` and LLMs subjecting to natural perturbations.

| Victim | SQuAD 1.1 | SQuAD 2.0 |
|---|---|---|
| flan-t5-small | −0.69 | −0.64 |
| flan-t5-base | −0.91 | −1.32 |
| flan-t5-large | −0.77 | −1.13 |
| flan-t5-xl | −0.98 | −1.37 |
| gemma-2-2b-it | − | −0.76 |
| gemma-2-9b-it | −0.89 | −0.92 |
| llama-3.1-8B-instruct | −0.38 | 0.39 |
| llama-3.2-3B-instruct | −0.96 | −0.37 |
| mistral-7B-instruct-v0.2 | 0.39 | −1.28 |
| falcon-7b-instruct | −0.88 | −5.38 |
| falcon-40b-instruct | −0.80 | − |

Afterwards, we also comprehensively measure the transferability of adversarial examples across all models and observe that these models exhibit similar error patterns, with LLMs (especially `Falcon`) showing moderate differences. However, the lowest transferability metric is still as high as 0.86.

## F  HUMAN ANNOTATION INSTRUCTIONS

In Figure 5, we show the instructions given to human annotators for error analysis (Section 5.2) and adversarial validity checking (Section 5.3), respectively. All our human annotators are students from universities in the United Kingdom and China. Before commencing each task, we ask them to annotate some examples and report the average time spent on each. As compensation, annotators receive 40 pence for each annotated example.

---

**Error Analysis**

You will be presented with pairs of reading contexts and their modified versions. The task is to compare each context and its modified version, observe the changes made and classify them into one or more of the semantic edit intention categories detailed below:

- *Copy Editing*: Rephrase; improve grammar, spelling, tone, or punctuation
- *Clarification*: Specify or explain an existing fact or meaning by example or discussion without adding new information
- *Elaboration*: Extend/add new content; insert a fact or new meaningful assertion
- *Fact Update*: Update numbers, dates, scores, episodes, status, etc. based on newly available information
- *Refactoring*: Restructure the article; move and rewrite content, without changing the meaning of it
- *Simplification*: Reduce the complexity or breadth of discussion; may remove information
- *Vandalism*: Deliberately attempt to damage the article
- *Other*: None of the above

We will use your annotation to calculate the percentage of each edit category.

**Adversarial Validity Checking**

Please read each provided context carefully and answer a corresponding question. Select the shortest continuous span from the context as your answer. If you believe a question cannot be answered, leave the answer blank. Your answer will be compared with the ground truth answers, and the result will only be used to decide the human answerability of the question.

---

Figure 5: Instructions for the two distinct human annotation tasks. In the error analysis task, the eight semantic edit intentions are adopted from (Yang et al., 2017).

## G  DEMONSTRATION OF PERTURBED MRC EXAMPLES FOR ENCODER-ONLY MODELS

Figure 6 illustrates a naturally perturbed MRC instance each for categories C2W, C2C, and W2C, with the annotated perturbation type(s).

## H  DETAILED EXPLANATION OF CHALLENGING TEST SET CONSTRUCTION

In Section 5.4, our aim is to zoom in on the errors of encoder-only models as much as possible and examine whether these errors transfer to Flan-T5 and LLMs. Therefore, we propose an exhaustive search algorithm to create the challenging natural perturbed test set:

Given a matched reading passage ( P ) from the prior version, its counterpart ( P' ) from the current version, and the associated questions:

**First Scenario**: We treat ( P ) as the original passage and ( P' ) as the perturbed one. We then evaluate, for each associated question, how many encoder-only models demonstrate the lack of robustness phenomenon, i.e., succeed on ( P ) but fail on ( P' ). We finally obtain the total number of models that demonstrate the lack of robustness phenomenon across all questions, denoted as ( N ). Questions on which none of the models demonstrate the lack of robustness phenomenon are removed, leaving ( Q ) questions.

---

**Category: C2W**

**Original Paragraph:** *Jacksonville, like most large cities in the United States, suffered from negative effects of rapid urban sprawl after World War II. The construction of highways led residents to move to newer housing in the suburbs. After World War II, the government of the city of Jacksonville began to increase spending to fund new public building projects in the boom that occurred after the war.* [...]

**Perturbed Paragraph:** *Jacksonville, like most large cities in the United States, suffered from negative effects of rapid urban sprawl after World War V. The construction of highways led residents to move to newer housing in the suburbs. After World War II, the government of the city of Jacksonville began to increase spending to fund new public building projects in the boom that occurred after the war.* [...]

**Question:** What did Jacksonville suffer from following World War I?

**Prediction of `distilbert-base` and `spanbert-large-cased`:** unanswerable→rapid urban sprawl

**Annotated Natural Perturbation Type:** Vandalism

---

**Category: C2C**

**Original Paragraph:** *Construction projects can suffer from preventable financial problems. Underbids happen when builders ask for too little money to complete the project. Cash flow problems exist when the present amount of funding cannot cover the current costs for labour and materials, and because they are a matter of having sufficient funds at a specific time, can arise even when the overall total is enough. Fraud is a problem in many fields, but is notoriously prevalent in the construction field. Financial planning for the project is intended to ensure that a solid plan with adequate safeguards and contingency plans are in place before the project is started and is required to ensure that the plan is properly executed over the life of the project.*

**Perturbed Paragraph:** *Financial planning ensures adequate safeguards and contingency plans are in place before the project is started, and ensures that the plan is properly executed over the life of the project. Construction projects can suffer from preventable financial problems. Underbids happen when builders ask for too little money to complete the project. Cash flow problems exist when the present amount of funding cannot cover the current costs for labour and materials; such problems may arise even when the overall budget is adequate, presenting a temporary issue. Fraud is also an occasional construction issue.*

**Question:** What can construction projects suffer from?

**Prediction of all encoder-only models:** preventable financial problems→preventable financial problems

**Annotated Natural Perturbation Type:** Copy Editing; Refactoring; Simplification

---

**Category: W2C**

**Original Paragraph:** *[...] The antigens expressed by tumors have several sources; some are derived from oncogenic viruses like human papillomavirus, which causes cervical cancer, while others are the organism's own proteins that occur at low levels in normal cells but reach high levels in tumor cells. [...] A third possible source of tumor antigens are proteins normally important for regulating cell growth and survival, that commonly mutate into cancer inducing molecules called oncogenes.*

**Perturbed Paragraph:** *[...] The antigens expressed by tumors have several sources; some are derived from oncogenic viruses like human papillomavirus, which causes cancer of the cervix, vulva, vagina, penis, anus, mouth, and throat,while others are the organism's own proteins that occur at low levels in normal cells but reach high levels in tumor cells. [...] A third possible source of tumor antigens are proteins normally important for regulating cell growth and survival, that commonly mutate into cancer inducing molecules called oncogenes.*

**Question:** What is a fourth possible source for tumor antigens?

**Prediction of `bert-base-uncased`:** proteins normally important for regulating cell growth and survival→unanswerable

**Annotated Natural Perturbation Type:** Elaboration

---

Figure 6: Natural perturbed MRC example in C2W, C2C and W2C categories.

**Second Scenario**: We treat ( P' ) as the original passage and ( P ) as the perturbed one. We then repeat the same evaluation process as described in the first scenario and obtain the total number of

models demonstrating the lack of robustness phenomenon across all questions, denoted as ( N' ). Questions on which none of the models demonstrate the lack of robustness phenomenon are removed as well, leaving ( Q' ) questions.

If ( $N > N'$ ), we consider ( P ) as the original passage and ( P' ) as the perturbed version.

If ( $N < N'$ ), we consider ( P' ) as the original and ( P ) as the perturbed.

If ( $N = N'$ ), we compare ( Q ) and ( Q' ):

- If ( $Q > Q'$ ), we consider ( P ) as the original passage and ( P' ) as the perturbed version.

- If ( $Q < Q'$ ), we consider ( P' ) as the original and ( P ) as the perturbed.

- If ( $Q = Q'$ ), the order does not matter, and we randomly decide which one should be the original and which should be the perturbed.

## I   NATURAL ADVERSARIAL SAMPLES FOR LLMS

We demonstrate two naturally perturbed reading comprehension examples that pose challenges for LLMs in Figure 7.

## J   SYNTHETIC PERTURBATION METHODS

Table 8 presents the synthetic perturbation methods used in this study.

Table 8: Various synthetic perturbation approaches.

| Method | Description |
|---|---|
| *character-level* | |
| CharOCR | Replace characters with Optical Character Recognition (OCR) errors. |
| CharInsert | Inject new characters randomly. |
| CharSubstitute | Substitute original characters randomly. |
| CharSwapMid | Swap adjacent characters within words randomly, excluding the first and last character. |
| CharSwapRand | Swap characters randomly without constraint. |
| *word-level* | |
| WInsert (CWE) | Insert new words to random position according to contextual word embeddings calculation from `RoBERTa-base` (Liu et al., 2019). |
| WSubstitute (CWE) | Substitute words according to contextual word embeddings calculation from `RoBERTa-base` (Liu et al., 2019). |
| WSplit | Split words to two tokens randomly. |
| WSwap | Swap adjacent words randomly. |
| WDelete | Delete words randomly. |
| WCrop | Remove a set of continuous word randomly. |
| Word Synonym Substitution (WSynSub) | Substitute words with synonyms from large size English PPDB (Pavlick et al., 2015). |
| WInsert (WE) | Insert new words to random position according to GloVe (Pennington et al., 2014) word embeddings calculation (we use *glove.6B.300d.txt*). |

We employ methods including WSplit, WSynSub and WInsert (WE) to each sentence in the original reading passage, and then recombine the modified sentences to generate the perturbed version. Conversely, other perturbation approaches are directly executed on the entire paragraph, as implementing them at the sentence-level might result in perturbed text that is even difficult for humans to read and comprehend (Si et al., 2021). The implementation of all character-level and word-level methods is carried out using the NLPAug library (Ma, 2019). Moreover, we set the perturbation rate to 30%, in line with the default settings within the NLPAug library.

---

**NAT_V1_CHALLENGE**

**Original Paragraph:** *In business, notable alumni include Microsoft CEO Satya Nadella, Oracle Corporation founder and the third richest man in America Larry Ellison, Goldman Sachs and MF Global CEO as well as former Governor of New Jersey Jon Corzine, McKinsey & Company founder and author of the first management accounting textbook James O. McKinsey, Arley D. Cathey, Bloomberg L.P. CEO Daniel Doctoroff, Credit Suisse CEO Brady Dougan, Morningstar, Inc. founder and CEO Joe Mansueto, Chicago Cubs owner and chairman Thomas S. Ricketts, and NBA commissioner Adam Silver.*

**Perturbed Paragraph:** *In business, notable alumni include Microsoft CEO Satya Nadella, Oracle Corporation founder and the third richest man in America Larry Ellison, Goldman Sachs and MF Global CEO as well as former Governor of New Jersey Jon Corzine, McKinsey & Company founder and author of the first management accounting textbook James O. McKinsey, co-founder of the Blackstone Group Peter G. Peterson, co-founder of AQR Capital Management Cliff Asness, founder of Dimensional Fund Advisors David Booth, founder of The Carlyle Group David Rubenstein, Lazard CEO Ken Jacobs, entrepreneur David O. Sacks, CEO of TPG Group and former COO of Goldman Sachs Jon Winkelreid, former COO of Goldman Sachs Andrew Alper, billionaire investor and founder of Oaktree Capital Management Howard Marks, Bloomberg L.P. CEO Daniel Doctoroff, Credit Suisse CEO Brady Dougan, Morningstar, Inc. founder and CEO Joe Mansueto, Chicago Cubs owner and chairman Thomas S. Ricketts, and NBA commissioner Adam Silver.*

**Question:** What Goldman Sachs CEO is also an alumni of the University of Chicago?

**Prediction of `GPT-3.5-turbo-0125` and `Llama-3-8B-Instruct`:** Jon Corzine→Jon Winkelreid

**Prediction of `Falcon-40B-Instruct`:** Jon Corzine→David Rubenstein, co-founder of The Carlyle Group, is also an alumnus of the University of Chicago.

---

**NAT_V2_CHALLENGE**

**Original Paragraph:** *Each chapter has a number of authors who are responsible for writing and editing the material. A chapter typically has two "coordinating lead authors", ten to fifteen "lead authors", and a somewhat larger number of "contributing authors". The coordinating lead authors are responsible for assembling the contributions of the other authors, ensuring that they meet stylistic and formatting requirements, and reporting to the Working Group chairs. Lead authors are responsible for writing sections of chapters. Contributing authors prepare text, graphs or data for inclusion by the lead authors.*

**Perturbed Paragraph:** *Each chapter has a number of authors to write and edit the material. A typical chapter has two coordinating lead authors, ten to fifteen lead authors and a larger number of contributing authors. The coordinating lead authors assemble the contributions of the other authors. They ensure that contributions meet stylistic and formatting requirements. They report to the Working Group co-chairs. Lead authors write sections of chapters. They invite contributing authors to prepare text, graphs or data for inclusion.*

**Question:** Who has the responsibility for publishing materials?

**Prediction of `Mistral-7B-Instruct-v0.2`:** Unanswerable. The text does not mention any responsibility related to publishing materials.→The coordinating lead authors are responsible for publishing materials in the given context.

---

Figure 7: Natural perturbed MRC examples that confuse LLMs.

## K   IMPACT OF SYNTHETIC ADVERSARIAL TRAINING

Figure 8 describes the impact of synthetic adversarial training (for `deberta-large`) on handling natural and synthetic perturbations.

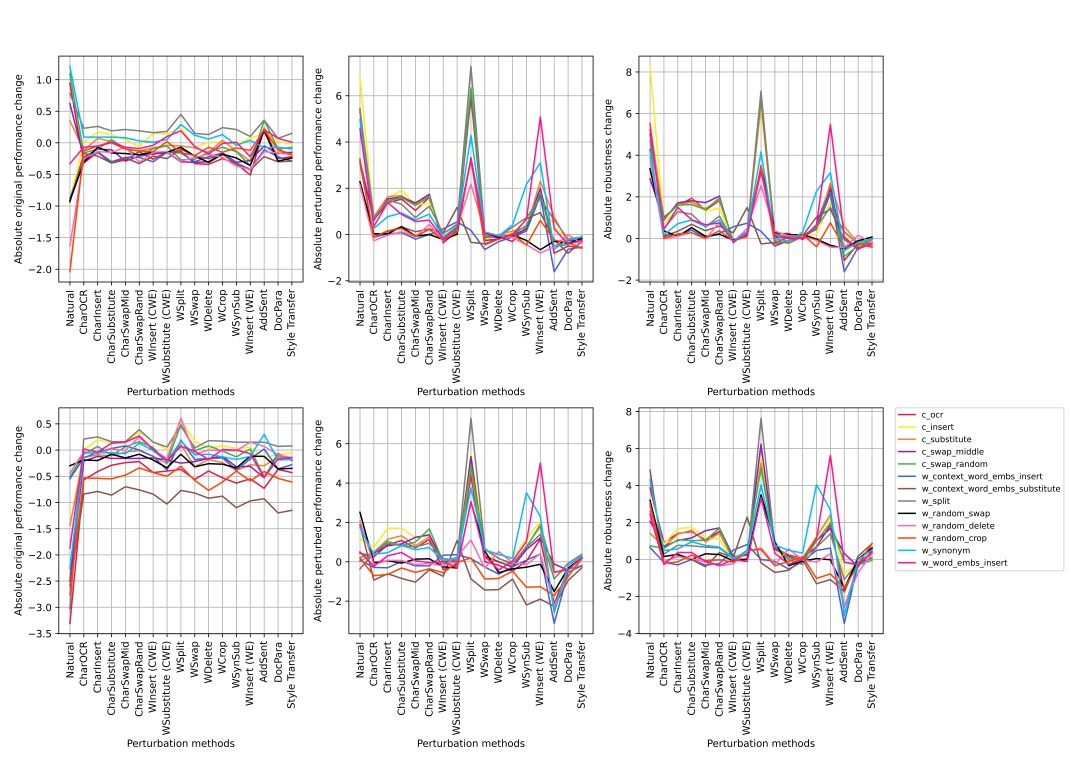

Figure 8: Absolute changes in original and perturbed performance (F1), as well as the robustness of `deberta-large` under natural and various synthetic noises, following retraining with each synthetic perturbation. The upper row and the bottom row illustrate the results on the SQUAD 1.1 and SQUAD 2.0 format test sets, respectively.

