# OpenReview forum: "Pay Attention to Real World Perturbations! Natural Robustness Evaluation in Machine Reading Comprehension"
_ICLR.cc/2025/Conference — Submitted to ICLR 2025_

### Official Review · Reviewer_fAbR · 2024-10-31

**Soundness:** 2
**Presentation:** 3
**Contribution:** 2
**Rating:** 3
**Confidence:** 4

**Summary:**

The authors study the impact of real world perturbations on MRC performance for various Transformer-based architectures. They propose a framework to create naturally-perturbed test sets from popular QA datasets using the revision history of Wikipedia. They perform experiments and analyses on the abovementioned models, followed by adversarial training of encoder-only models.

**Strengths:**

* Comprehensive evaluation across multiple architecture types rather than only decoder models and multiple models of each type: encoder, decoder, encoder-decoder models
* Somewhat comprehensive evaluation across multiple QA datasets, with caveats: see below
* Paragraphs are generally well-written and easy to understand

**Weaknesses:**

**TLDR**

The authors pose an interesting question, but the execution of the study contains unexpected design decisions that are not well-justified. The exact improvement of their claimed methodology over existing work is also unclear.

**Details:**
* The authors call out the similarities between their method and Belinkov & Bisk (2018), do not make clear the differences and improvements over the latter, if any. The claimed contribution ("novel Wikipedia revision history-based framework") also does not highlight the exact improvements over simply applying Belinkov & Bisk (2018)'s method to the MRC setting, and is therefore running the risk of over-claiming. I recommend the authors revise the framing to highlight the exact contribution over Belinkov & Bisk (2018) and other prior, similar work [1, 2].
* The encoder-only models were only evaluated on SQuAD datasets to draw conclusions, with no clear justification even though the authors are aware that they are no longer challenging benchmarks.
* The transition to experiments on the other architectures and datasets was a little strange (only evaluating on questions that the encoder models failed on) with no clear justification. There is then a transition back to encoder-only models and SQuAD for adversarial training, also without clear justification for excluding other datasets and architectures.
* The authors appear to have missed relevant work on robustness evaluation in QA/MRC [3, 4] and other types of synthetic perturbations [5].


**Missing References:**
1. Mining Naturally-occurring Corrections and Paraphrases from Wikipedia’s Revision History. Max et al. 2010.
1. Robust Systems for Preposition Error Correction Using Wikipedia Revisions. Cahill et al. 2013.
1. It’s Morphin’ Time! Combating Linguistic Discrimination with Inflectional Perturbations. Tan et al. 2020.
1. Code-Mixing on Sesame Street: Dawn of the Adversarial Polyglots. Tan et al. 2021.
1. From Hero to Zéroe: A Benchmark of Low-Level Adversarial Attacks. Eger et al 2020.

**Questions:**

> Questions on which none of the encoder-only models fail under the perturbation are then removed.

Why was this decision made, rather than studying the effect of all perturbed questions on each architecture type? An analysis of the overlaps between architectures is good to have, but I would have preferred to see the former if I had to choose one. I may have missed a strong justification, if one has already been included.

---

> ### Author Response · Authors · 2024-11-16
> **Address weakness 1 and 2**
>
> Dear reviewer,
>
> We are very grateful for your efforts in reviewing our work and proposing these concerns. In the following, we address the raised concerns and look forward to further discussion if there is anything unclear.
>
> >Weakness 1: The authors call out the similarities between their method and Belinkov & Bisk (2018), do not make clear the differences and improvements over the latter, if any. The claimed contribution ("novel Wikipedia revision history-based framework") also does not highlight the exact improvements over simply applying Belinkov & Bisk (2018)'s method to the MRC setting, and is therefore running the risk of over-claiming. I recommend the authors revise the framing to highlight the exact contribution over Belinkov & Bisk (2018) and other prior, similar work.
>
> Many thanks for pointing this out! Our approach to construct natural perturbations is inspired by (Belinkov & Bisk, 2018) [1]. While we all use Wikipedia revision histories as the source of natural perturbations, the major difference is that:
>
> Perturbation in [1] is restricted to single word replacements and applied on non-English source-side sentences in machine translation. In detail, they build a look-up table of possible lexical replacements by harvesting naturally occurring errors (typos, misspellings, etc.) from available corpora of French/German Wikipedia edits [2, 3]. Afterwards, they replace every word in the source-side sentences with an error if one exists in the look-up table. Different from [1], our approach does not restrict the perturbation level and utilise English Wikipedia. We replace the whole reading passage with the edited version available in English Wikipedia revision history. This enables us to capture more comprehensive and critical natural perturbation patterns (see Section 5.2) that can not be possible to capture in [1].
>
> Also, as the reviewer mentioned, there are a series of works involving the information extraction from Wikipedia revisions. However, their aim is to investigate the benefits of such extracted data for specific tasks like spelling correction [2, 3], grammatical error correction [4] and lexical simplification [5], rather than for robustness assessment. Therefore, compared to our work, these studies may overlook certain natural perturbation patterns that are critical in robustness evaluation settings.
>
> Overall, we believe our claimed contribution is valid. We will further support it by highlighting the differences and improvements of our approach over previous works [1, 2, 3, 4, 5]. Many thanks again to the reviewer for this valuable question!
>
> [1] Yonatan Belinkov and Yonatan Bisk. Synthetic and natural noise both break neural machine translation. In ICLR, 2018.
> [2] Aurlien Max and Guillaume Wisniewski. Mining Naturally-occurring Corrections and Paraphrases from Wikipedias Revision History. In LREC, 2010.
> [3] Torsten Zesch. Measuring Contextual Fitness Using Error Contexts Extracted from the Wikipedia Revision History. In EACL, 2012.
> [4] Aoife Cahill, Nitin Madnani, Joel Tetreault, and Diane Napolitano. Robust systems for preposition error correction using wikipedia revisions. In NAACL, 2013.
> [5] Mark Yatskar, Bo Pang, Cristian Danescu-Niculescu-Mizil, and Lillian Lee. For the sake of simplicity: Unsupervised extraction of lexical simplifications from Wikipedia. In NAACL, 2010.
>
> >Weakness 2: The authors appear to have missed relevant work on robustness evaluation in QA/MRC and other types of synthetic perturbations.
>
> Many thanks for reminding us of these valuable related works, we will add them to our paper.

---

> ### Author Response · Authors · 2024-11-16
> **Address weakness 3**
>
> We appreciate the reviewer’s detailed feedback. All the concerns in weakness 3 are about the experimental design justification, therefore we address them together in the order of the analyses presented in Section 5.
>
> > The encoder-only models were only evaluated on SQuAD datasets to draw conclusions, with no clear justification even though the authors are aware that they are no longer challenging benchmarks.
>
> Our intention in starting with SQuAD and encoder-only models was to establish a baseline evaluation of model behaviour under natural perturbations. While SQuAD is less challenging, its simplicity enables a focused and controlled examination of the perturbation effects (Section 5.1), error sources (Section 5.2) and adversarial instance validity (Section 5.3), providing a foundation for generalising our findings to more complex datasets and model architectures.
>
> > The transition to experiments on the other architectures and datasets was a little strange (only evaluating on questions that the encoder models failed on) with no clear justification. ("Questions on which none of the encoder-only models fail under the perturbation are then removed.") -- Why was this decision made, rather than studying the effect of all perturbed questions on each architecture type? An analysis of the overlaps between architectures is good to have, but I would have preferred to see the former if I had to choose one. I may have missed a strong justification, if one has already been included.
>
> As we observe that encoder-only models suffer from natural perturbations on the SQuAD datasets, we further ***investigate the transferability of the errors generated by encoder-only models to other model architectures***—a common approach in robustness evaluation research [1]. To do so, we zoom in on the errors of encoder-only models, removing questions on which none of the encoder models fail (since they do not fail the encoder-only models), and evaluate the performance change of Flan-T5 and LLMs on the collected adversarial examples (see Table 2 in Section 5.4 for the results).
>
> We agree with the reviewer that studying the effect of all perturbed instances on each architecture type is also valuable. We are currently running experiments on this and will update the results in the coming days.
>
> We also evidence that the behaviour we observed in the baseline evaluation in Section 5.1 (i.e., encoder-only models suffer from natural perturbations on the SQuAD datasets) also carries over to more powerful LLMs and other more complex datasets (see Table 3 in Section 5.5 for the results).
>
> [1] Robin Jia and Percy Liang. Adversarial Examples for Evaluating Reading Comprehension Systems. In EMNLP, 2017.
>
> > There is then a transition back to encoder-only models and SQuAD for adversarial training, also without clear justification for excluding other datasets and architectures.
>
> For adversarial training:
> 1. We focused on encoder-only models and the SQuAD datasets to maintain consistency with our baseline evaluations in Section 5.1. Our work shows that while adversarial training can help encoder-only models address natural perturbations, it falls far short of effectively mitigating them, warranting future study.
> 2. Expanding to other datasets and architectures could be done in future work. Our aim/contribution in this paper is more to raise awareness within the community about models’ vulnerabilities to this quite under-explored natural perturbations rather than mitigate it.

---

> > ### Author Response · Authors · 2024-11-20
> > **Supplementary experiments**
> >
> > As suggested by the reviewer, we supplemented Table 1 in Section 5.1 with additional experiments on Flan-T5 and more recent LLMs such as Gemma 2 and Llama 3.2, to study the effect of all perturbed instances on each architecture type. The results are presented in Table R. From Table R, we observe that similar to encoder-only models, Flan-T5 and LLMs generally exhibit varying degrees of performance degradation under natural perturbations, but also exhibit considerable robustness to them.
> >
> > Table R: Performance change (%) for Flan-T5 and LLMs subjecting to natural perturbations.
> > | **Victim**                | **SQuAD 1.1** | **SQuAD 2.0** |
> > |---------------------------|---------------|---------------|
> > | flan-t5-small             | -0.69         | -0.64         |
> > | flan-t5-base              | -0.91         | -1.32         |
> > | flan-t5-large             | -0.77         | -1.13         |
> > | flan-t5-xl                | -0.98         | -1.37         |
> > | gemma-2-2b-it             | -             | -0.76         |
> > | gemma-2-9b-it             | -0.89         | -0.92         |
> > | llama-3.1-8B-instruct     | -0.38         | 0.39          |
> > | llama-3.2-3B-instruct     | -0.96         | -0.37         |
> > | mistral-7B-instruct-v0.2  | 0.39          | -1.28         |
> > | falcon-7b-instruct        | -0.88         | -5.38         |
> > | falcon-40b-instruct       | -0.80         | -             |
> >
> > Afterwards, we also comprehensively measure the transferability of adversarial examples across all models and observe that these models exhibit similar error patterns, with LLMs (especially Falcon) showing moderate differences. However, the lowest transferability metric is still as high as 0.86.

---

> > > ### Comment · Reviewer_fAbR · 2024-11-21
> > > **Follow-up**
> > >
> > > Thanks for the detailed response! Is there a strong reason to not include the encoder-decoder and decoder-only  results as part of Table 1 and expand the error analysis to all architectures, as opposed to simply including these results in the appendix and focusing on transfer in the main text?
> > >
> > > > a common approach in robustness evaluation research [1]
> > >
> > > I don't follow this argument. The reference is from 2017, when classifier models were the dominant architecture/pretraining method in NLP. Generation models are currently the focus, so results/analysis on them would be equally, if not more, relevant to the community.

---

> ### Author Response · Authors · 2024-11-23
>
> Thank you for the follow-up comment.
>
> By referencing [1], we aim to emphasise the importance of investigating the transferability of adversarial examples. In line with the approach suggested by [2]—which advocates for re-evaluating claims with modern techniques—we chose to examine how errors observed in earlier encoder-only models may persist in SOTA LLMs, which are now central to NLP research. This is exactly what the reviewer expects. This approach is both fair and meaningful, as we utilise encoder-only architecture for the creation of challenging test sets in Section 5.4, which contrasts significantly with the generative LLMs we evaluate. Such model-in-the-loop challenge set creation approach, as seen in [3], is also a common practice and is central to our methodology.
>
> Precisely the generative LLMs is what we evaluate. By identifying errors made by BERT-style models, we assess whether these same errors are exhibited by LLMs. Our results show that, while these errors appear benign to human annotators, they largely transfer to LLMs, revealing weaknesses in the current generation of LLMs under natural perturbations.
>
> The investigation into the effect of all perturbed instances on Flan-T5 and other LLMs, as presented in Table R, is supplementary and serves to broaden the scope of our experimentation. For clarity and conciseness, we have included this additional analysis in Appendix E.
>
> We hope this clarification aligns with your expectations and enhances the understanding of our approach.
>
> [1] Robin Jia and Percy Liang. Adversarial Examples for Evaluating Reading Comprehension Systems. In EMNLP, 2017.
> [2] Samuel R. Bowman. The Dangers of Underclaiming: Reasons for Caution When Reporting How NLP Systems Fail. In ACL, 2022.
> [3] Naik et al. Stress Test Evaluation for Natural Language Inference. In COLING, 2018.

---

> > ### Comment · Reviewer_fAbR · 2024-11-26
> >
> > I disagree with the impact/relevance of this framing and keep my score.

---

### Official Review · Reviewer_qQoy · 2024-11-02

**Soundness:** 3
**Presentation:** 3
**Contribution:** 2
**Rating:** 3
**Confidence:** 4

**Summary:**

This paper introduces perturbed sets of several machine-reading comprehension datasets including SQuAD, using Wikipedia edits to create natural perturbations. Results show that encoder-only, decoder-only and encoder-decoder LMs suffer from this challenge. Adversarial training with perturbation is an effective defense strategy.

**Strengths:**

1. It is interesting to use Wikipedia edit history to construct perturbation.
2. The perturbed set is verified by human to ensure that the perturbed examples are still valid.
3. Results show that natural perturbation is a powerful attack to LMs.

**Weaknesses:**

1. It is unclear whether stronger model, e.g. gpt-4o would still suffer from this challenge. While weaker models like BERT suffers from the natural perturbations, it is important to show that it is still a challenge for recent stronger LLMs.
2. The perturbation method relies on Wikipedia edit history, limiting its applicability to non-Wikipedia based datasets.
3. The performance drops on non-SQuAD datasets like DROP are relatively small, e.g. LLaMA-2 only exhibits less than 2 points drop, which could also potentially be remedied by adversarial training. Again, I'm concerned that this benchmark is not super challenging for recent LLMs anymore.

**Questions:**

line 334-337: I have trouble understanding this long sentence with too many clauses. can you explain that?

---

> ### Author Response · Authors · 2024-11-14
> **Address weaknesses**
>
> We appreciate the reviewer for the time spent in reviewing our work and address the proposed concerns below.
>
> > It is unclear whether stronger model, e.g. gpt-4o would still suffer from this challenge. While weaker models like BERT suffers from the natural perturbations, it is important to show that it is still a challenge for recent stronger LLMs. The performance drops on non-SQuAD datasets like DROP are relatively small, e.g. LLaMA-2 only exhibits less than 2 points drop, which could also potentially be remedied by adversarial training. Again, I'm concerned that this benchmark is not super challenging for recent LLMs anymore.
>
> As one of the strengths recognised by the reviewer, our work show that natural perturbation is a powerful attack to not only encoder-only and Flan-T5 models, but also to the recent Large Language Models (LLMs), including Google’s Gemma (2B and 7B), Meta’s Llama 2 (7B and 13B) and Llama 3 (8B), Mistral 7B and Falcon (7B and 40B). To further evidence its harmfulness, we experimented with GPT-4o. Table 1 shows the results.
>
> #### Table 1: IM changes (%) of GPT-4o on naturally perturbed test sets of five MRC datasets.
> | Dataset  | SQuAD V1 | SQuAD V2 | DROP   | HotpotQA | BoolQ  |
> |----------|----------|----------|--------|----------|--------|
> | **GPT-4o** | -13.06   | -13.39   | -12.68 | -2.67    | -7.47  |
>
> It can be seen clearly from Table 1 that GPT-4o still suffers from natural perturbations, with quite noticeable performance drops. This further demonstrates the harmfulness of natural perturbations to the contemporary powerful LLMs.
>
> Also, we indeed notice that the performance decrease of some models on certain datasets (e.g., Llama 2 on DROP) is not that significant. However:
>
> 1. Our aim/contribution is more to raise awareness within the community about models’ vulnerabilities to the quite under-explored natural perturbations, rather than mitigate it.
> 2. The effectiveness of adversarial training for LLMs remains uncertain. Our work shows that while it can help encoder-only models address natural perturbations, it falls far short of effectively mitigating them, warranting future study.
>
> > The perturbation method relies on Wikipedia edit history, limiting its applicability to non-Wikipedia based datasets.
>
> In this paper, our goal is to show the behaviour of neural language models (e.g., LLMs) on natural perturbations. These are by no means limited to Wikipedia, but occur in any kind of text that evolves over time, e.g., company-internal files, software documentation, etc. Wikipedia happens to allow us to track changes and automatically construct a benchmark to test the behaviour, but the phenomenon of natural perturbations is by no means limited to Wikipedia. Our work serves as an initial step to address this critical challenge, and future work is needed to explore alternative natural perturbation approaches, as mentioned in Conclusion.

---

> > ### Comment · Reviewer_qQoy · 2024-11-21
> > **Response**
> >
> > Thank you for your response to me review. Could you please clarify on which test set do you conduct experiments in Table 1? Do you still use the questions that the encoder models failed on? I agree with Reviewer fAbR that this setting is strange and not well justified. According to Table R in response to Reviewer fAbR, the performance drop on the whole test set is minimal, questioning the impact of such natural perturbations.

---

> ### Author Response · Authors · 2024-11-14
> **Answer question**
>
> > line 334-337: I have trouble understanding this long sentence with too many clauses. can you explain that?
>
> Yes, sure, and sorry for the confusion. In Section 5.4, our aim is to zoom in on the errors of encoder-only models as much as possible and examine whether these errors transfer to Flan-T5 and LLMs. Therefore, we propose an exhaustive search algorithm to create the challenging natural perturbed test set:
>
> Given a matched reading passage \( P \) from the prior version, its counterpart \( P' \) from the current version, and the associated questions:
>
> 1. **First Scenario:** We treat \( P \) as the original passage and \( P' \) as the perturbed one. We then evaluate, for each associated question, how many encoder-only models demonstrate the lack of robustness phenomenon, i.e., succeed on \( P \) but fail on \( P' \). We finally obtain the total number of models that demonstrate the lack of robustness phenomenon across all questions, denoted as \( N \). Questions on which none of the models demonstrate the lack of robustness phenomenon are removed, leaving \( Q \) questions.
>
> 2. **Second Scenario:** We treat \( P' \) as the original passage and \( P \) as the perturbed one. We then repeat the same evaluation process as described in the first scenario and obtain the total number of models demonstrating the lack of robustness phenomenon across all questions, denoted as \( N' \). Questions on which none of the models demonstrate the lack of robustness phenomenon are removed as well, leaving \( Q' \) questions.
>
> If \( N > N' \), we consider \( P \) as the original passage and \( P' \) as the perturbed version.
> If \( N < N' \), we consider \( P' \) as the original and \( P \) as the perturbed.
> If \( N = N' \), we compare \( Q \) and \( Q' \):
> - If \( Q > Q' \), we consider \( P \) as the original passage and \( P' \) as the perturbed version.
> - If \( Q < Q' \), we consider \( P' \) as the original and \( P \) as the perturbed.
> - If \( Q = Q' \), the order does not matter, and we randomly decide which one should be the original and which should be the perturbed.
>
> We hope our response addresses the reviewer’s concern, and we are happy to discuss if there is anything unclear.
>
> Many thanks in advance for your precious time to read our response and participate the discussion!

---

> > ### Comment · Reviewer_qQoy · 2024-11-21
> > **Response to line 334-337**
> >
> > Thank you for your clarification on line 334-337 of how you contracted the natural perturbed set. However, it is unclear to me why you don't simply treat the passages from prior version as original and those from current version as perturbed. Your proposed method seems to intentionally identify a "natural" test set that hacks the performance drop, which is an unnatural design to me.

---

> > > ### Author Response · Authors · 2024-11-28
> > >
> > > Dear Reviewer qQoy,
> > >
> > > As the discussion period draws to a close, could you please kindly update your review/score based on our clarifications and responses to your questions?
> > >
> > > Thank you for your consideration.

---

> ### Author Response · Authors · 2024-11-21
>
> Many thanks for your response!
>
> In Table 1 above, for the two SQuAD datasets, we evaluate GPT-4o on the instances on which encoder-only models failed on. This is because we are investigating whether the errors of encoder-only models carry over to GPT-4o as well. We provided a detailed justification for this setting in our response to Reviewer fAbR.
>
> In Table R, it is expected that the performance drop of more powerful Flan-T5 and LLMs on the "whole" test set is not significant, similar as what we observe for encoder-only models in Section 5.1. However, we later show that Flan-T5 and LLMs commit errors produced by encoder-only models significantly on two SQuAD datasets and also show noticeable performance decrease on other challenging datasets such as HotpotQA, indicating the harmfulness of such natural perturbations.

---

> ### Author Response · Authors · 2024-11-21
>
> Thank you for your question. We indeed applied the approach of simply treating the passages from the prior version as original and those from the current version as perturbed (as described in Section 3 line 155-157), generating the naturally perturbed "whole" test set used for evaluation in Section 5.1. However, this method resulted in non-significant performance drop (as you noticed in Table R), which limited the scope of errors we could observe.
>
> To address this limitation, we proposed an exhaustive search algorithm (Section 5.4) to zoom in on the errors made by encoder-only models and investigate whether these errors also manifest in Flan-T5 and LLMs. This method is not intended to artificially hack the performance drop but rather to create a more targeted and meaningful challenge set, to examine whether the types of errors seen in encoder-only models also persist in SOTA LLMs.
>
> Further, In this work, we use the term "natural" to refer to perturbations that occur genuinely in the real world. Their naturalness is not influenced by our experimental method but is inherent to the data itself.

---

### Official Review · Reviewer_6Syj · 2024-11-10

**Soundness:** 3
**Presentation:** 2
**Contribution:** 3
**Rating:** 6
**Confidence:** 3

**Summary:**

This paper contributes to understanding and improving the robustness of Machine Reading Comprehension models by introducing a new evaluation framework based on naturally occurring text variations. Rather than relying on synthetic perturbations, the authors leverage Wikipedia edit history to generate realistic test cases that reflect how the text changes in the real world.

**Strengths:**

- Proposes a framework using Wikipedia edit history to generate natural perturbations in MRC benchmarks

- Evaluate model performance across encoder-only, encoder-decoder, and decoder-only architectures

- Shows that natural perturbations can degrade performance and these errors transfer to larger models

- Demonstrate that adversarial training with both natural and synthetic perturbations can help mitigate these issues

**Weaknesses:**

I am generally optimistic about the paper, and I have the following minor concerns.

The analysis section could be more in-depth
1. Permutations and models

- No investigation of how perturbation magnitude affects performance and why certain permutations affect the model more than others;

- Missing analysis of the interaction between model size and robustness is extremely important as we see that some observations might not always be predictable and transferrable on smaller model sizes.

- I am wondering why authors didn't consider ablation studies on the impact of Wikipedia edit types.

2. Wikipedia edit history

It feels to me that Wikipedia's edit history might suffer from being less realistic and potentially data contamination.

**Questions:**

- why certain permutations affect the model more than others;

- Are certain types of changes over/under-represented?

- whether different pretraining approaches affect models' robustness to natural perturbations?

- the effectiveness of adversarial training vary with model size and architecture?

- the effectiveness of adversarial training vary with model size and architecture?

---

> ### Author Response · Authors · 2024-11-17
> **Address weaknesses**
>
> Thank you so much for taking the time to review our work. We greatly appreciate your insightful feedback and are delighted that you have such a positive impression of our paper. In the following, we address the raised weaknesses and questions, and look forward to further discussion.
>
> >No investigation of how perturbation magnitude affects performance and why certain permutations affect the model more than others
>
> We are currently investigating how perturbation magnitude affects performance and will update the results in the coming days.
>
> Figure 3 shows that among the 210 error cases of encoder-only models, *Copy Editing* (i.e., Rephrase; improve grammar, spelling, tone, or punctuation) contributes to more than 40%, while other perturbation types, such as *Clarification* and *Fact Update*, contribute less than 10%. This is likely because *Copy Editing* more frequently alters the answer sentences (i.e., the sentences required to answer the question) in the reading passage, which is the primary cause [1] of the models' confusion.
>
> [1] We show this empirically. Please check line 299-305.
>
> >Missing analysis of the interaction between model size and robustness is extremely important as we see that some observations might not always be predictable and transferrable on smaller model sizes.
>
> Our results demonstrate that the robustness of models under natural perturbations does not necessarily correlate with their size. For example, on NAT_V2_CHALLENGE, Llama 2-chat-7B shows overall higher robustness than Llama 2-chat-13B; flan-t5-xl features the largest performance decrease (12.79%) compared to its small and large versions.
>
> We will add a detailed analysis to the paper.
>
> >I am wondering why authors didn't consider ablation studies on the impact of Wikipedia edit types.
>
> This is because one revision could contain multiple perturbation types, making it challenging to isolate the individual effects. Additionally, automatically and precisely identifying these perturbation types remains a significant challenge.
>
> Despite this, our analysis in Section 5.2 provides a meaningful approximation. Out of 210 annotated C2C and C2W examples, approximately 86.7% and 81.4% involve only one perturbation type, respectively. This allows us to largely mitigate the influence of samples annotated with multiple perturbation types in the analysis. By examining the distribution of perturbation types across C2C and C2W examples (Figure 3), we observe that *Copy Editing* is the most frequent and impactful type that confuses models, while other types have a lesser effect.
>
> We will further explore more comprehensive ablation studies in future work.
>
> >It feels to me that Wikipedia's edit history might suffer from being less realistic and potentially data contamination.
>
> Thanks for this question! We would appreciate more detailed insights into why Wikipedia’s revision history might be considered less realistic.
>
> Regarding data contamination, this is indeed a critical issue in the NLP community today. Investigating potential data contamination and its impact remains a significant challenge, and we view it as a valuable direction for future work.

---

> ### Author Response · Authors · 2024-11-17
> **Answer questions**
>
> >Are certain types of changes over/under-represented?
>
> As shown in Figure 3, perturbation types *Copy Editing*, *Elaboration* appears more frequently than others such as *Clarification*, *Fact Update* and *Refactoring*. This distribution in general aligns with the edit intentions distribution annotated in previous work [1], and we believe it reflects the inherent nature of Wikipedia revisions.
>
> [1] Diyi Yang, Aaron Halfaker, Robert Kraut, and Eduard Hovy. Identifying semantic edit intentions from revisions in wikipedia. In EMNLP, 2017.
>
> >whether different pretraining approaches affect models' robustness to natural perturbations?
>
> We are currently running experiments on this and will update the results in the coming days.
>
> >the effectiveness of adversarial training vary with model size and architecture?
>
> Thank you for bringing this up! Yes, as shown in Table 4 and Figure 4, the effectiveness of adversarial training indeed varies with model size and architecture. Our results demonstrate that in general, adversarial training yields the most significant improvements in performance on the perturbed test set and overall robustness for the smallest evaluated model, distilbert-base. However, the benefits diminish in larger and more complex models.
>
> We would like to thank you again for your thoughtful questions. Please do not hesitate to let us know if anything remains unclear. Many thanks in advance for your time and effort during the discussion phase!

---

### Author Response · Authors · 2024-11-21
**Changes to our manuscript**

Dear Reviewers,

We again thank you for your efforts in reviewing our work and providing insightful feedback. Based on the weaknesses and questions raised by each reviewer, we made the following changes in the updated manuscript (changes are marked in blue):

**(Reviewer 6Syj)**
1. Investigated how perturbation magnitude affects performance *(Section 5.2 footnote 8)* and further clarified why certain permutations affect the model more than others *(Section 5.2 line 343-345)*.
2. Added analysis of the interaction between model size and robustness. *(Section 5.4 396-400)*
3. Further discussed the overall distribution of perturbation types *(Section 5.2 327-329)* and the effectiveness of adversarial training *(Section 6 481-484)*.

**(Reviewer qQoy)**
1. Added experimental results on GPT-4o. *(Section 4.2 227; Table 2 414; Table 3 447-448)*
2. Explained the applicability limitation in this work. *(Section 1 footnote 1)*
3. Described the challenge test set construction process in detail. *(Section 5.4 381-382; Appendix H)*

**(Reviewer fAbR)**

1. Highlighted the differences and improvements of our natural perturbation construction methodology over previous works. *(Section 1 075-081 and 083-085)*
2. Referenced the suggested related works. *(Section 1 045; Section 2 116-117)*
3. Added experimental design justification. *(Section 5 247-256; Section 6 455-457)*
4. Presented supplementary experiments to study the effect of all perturbed instances on each architecture type. *(Appendix E)*

---

### Meta-Review · Area_Chair_i23H · 2024-12-25

**Metareview:**

The paper introduces a new framework for evaluating the robustness of Machine Reading Comprehension (MRC) models by leveraging Wikipedia edit history -- naturally occurring edits to construct test sets that reflect real-world text variations. The paper evaluates how these naturally perturbed examples affect various Transformer architectures (encoder-only, encoder-decoder, and decoder-only) on multiple MRC datasets. In addition, the authors explore adversarial training with naturally perturbed data to help reduce accuracy drops.

**Strengths** (1) Using real-world perturbations is interesting and technically sound, complementing synthetic perturbations. (2) the study covers multiple architectures, which shows the breadth of evaluation.

**Weaknesses** (1) Reviewers highlight the insufficient analysis of various perturbation types, model sizes, and more challenging datasets & tasks; (2) The paper's experimental design for different architectures appears inconsistent as raised by some reviewers; (3) A deeper discussion of potential data contamination would strengthen the paper


**Decision** The paper received mixed reviews, and the author also addressed part of the questions of reviewers. However, while all reviewers agree the paper targets the important area of real-world robustness, there still limitations in the current form which the reviewers cannot agree on after the rebuttal. Therefore, I am leaning toward rejection.

**Additional Comments On Reviewer Discussion:**

The authors responded to the reviewers’ comments, and all three reviewers subsequently provided follow-up feedback.

However, both fAbR and qQoy ultimately chose not to change their original scores.

fAbR offered only a brief justification—“I disagree with the impact/relevance of this framing and will keep my score”—which provides little substance for the final decision. Meanwhile, qQoy remains unconvinced by the study’s design. As a result, I believe the manuscript may still not be ready for acceptance.

---

### Decision · Program_Chairs · 2025-01-22

Reject